# Clinical and Nutritional Effectiveness of a Nutritional Protocol with Oligomeric Enteral Nutrition in Patients with Oncology Treatment-Related Diarrhea

**DOI:** 10.3390/nu12051534

**Published:** 2020-05-25

**Authors:** Alejandro Sanz-Paris, Javier Martinez-Trufero, Julio Lambea-Sorrosal, Fernando Calvo-Gracia, Raimon Milà-Villarroel

**Affiliations:** 1Department of Endocrinology and Nutrition, Miguel Servet Hospital, 50009 Zaragoza, Spain; 2Instituto de Investigación Sanitaria Aragón (IIS Aragon), 50009 Zaragoza, Spain; 3Department of Oncology, Miguel Servet Hospital, 50009 Zaragoza, Spain; jmtrufero@seom.org; 4Department of Oncology, University Clinic Hospital, 50009 Zaragoza, Spain; juliolambea@yahoo.es; 5Department of Endocrinology and Nutrition, University Clinic Hospital, 50009 Zaragoza, Spain; fcalvo@comz.org; 6Group Research on Wellbeing (GRoW), Blanquerna School of Health Sciences–Universitat Ramon Llull, 08025 Barcelona, Spain; raimonmv@blanquerna.url.edu

**Keywords:** oncology treatment-related diarrhea, nutritional protocol, oligomeric enteral nutrition

## Abstract

(1) Background: Poor nutritional status and diarrhea are common complications in cancer patients. (2) Methods: This multicenter, observational, prospective study evaluated the effectiveness of an oligomeric enteral nutrition (OEN) protocol in the improvement of nutritional status and reduction of diarrhea symptoms. Nutritional status was assessed with the Subjective Global Assessment (SGA), Body Mass Index (BMI) and albumin levels. Diarrhea was evaluated by the frequency and consistency of stools (Bristol Stool form scale). (3) Results: After 8 weeks of OEN protocol, the nutritional status improved in 48.3% of patients, with an increased proportion of patients at risk of malnourishment (+27.3%) at the expense of a decrease of moderately (−19.9%) and severely (−7.3%) malnourished patients (*p* < 0.001). Serum albumin and BMI significantly increased after 8 weeks of OEN treatment (*p* < 0.005). OEN showed a 71.1% effectiveness in the improvement of stool consistency. The mean number of stools per day significantly decreased from baseline (4.17 stools/day) to week 8 (1.42 stools/day; *p* = 0.0041). The nutritional status significantly improved even in those patients with persistent diarrhea. (4) Conclusion: The proposed OEN protocol seemed to be effective in improving the nutritional status, frequency and consistency of stools in patients with oncology treatment-related diarrhea even in persistent cases.

## 1. Introduction

Despite the improvement in patient’s prognosis and life expectancy enabled by cancer therapy, oncological patients frequently suffer from associated side effects, being gastrointestinal complications particularly common [1].

Chemotherapy, other systemic treatments, and radiotherapy can induce and exacerbate the inflammation of the gastrointestinal tract mucosa, resulting in taste changes, nausea, constipation and diarrhea [2]. Of particular importance is oncology treatment-related diarrhea (OTRD) due to its effects on hydration and electrolyte balance, which can severely compromise the nutritional status of patients [3]. The frequency of OTRD ranged from 5 to 74% in randomized clinical trials, becoming a common cause of hospital admissions and being life-threatening in persistent and severe cases [4].

Cancer patients are therefore at high risk for malnourishment, with a prevalence that varies widely with the type and stage of tumor, cancer therapy modality and intensity [5]. Previous studies showed that poor nutritional status was associated with a higher rate of hospital admissions, increased hospital length of stay, reduced quality of life and mortality, and lower tolerance to therapy in cancer patients [6]. All these complications add an extra burden to the negative effects that cancer itself has for patients and healthcare systems [7].

Since these complications are progressive, early identification and intervention in patients at risk will minimize the progression towards more severe states that could prevent from continuing cancer therapy and threaten the life of the patient [7]. However, clinical guidelines mostly focus on the pharmacological treatment of diarrhea but do not specifically address the required nutritional support for patients [4,5,6,8,9,10]. Depending on the severity and persistence of symptoms, these interventions can range from dietary counselling and food supplements to medical nutrition therapy [11]. Among the latter, oral enteral nutrition can be an advantageous alternative to meet the nutritional demands of patients with OTRD and clinical stability [12].

However, specific guidance on this type of nutritional support is scarce. To address this unmet need, we recently published a nutritional support algorithm for patients with OTRD using oligomeric enteral nutrition (OEN) [13]. OEN is a nutritionally complete formula in which peptides (instead of whole proteins) are the source of nitrogen and fats are mainly provided as medium-chain triglycerides. This readily digestible source of nutrients increases digestibility, thereby facilitating the absorption of nutrients, protecting mucosal integrity, and showing low antigenic effect [14]. We developed a stepwise approach for patients with OTRD that considers the nutritional status (at risk of malnourishment, moderate or severe malnourishment) and the intestinal function (preserved or impaired) of patients. Depending on the severity and persistence of symptoms, the OEN protocol is recommended in different stages of OTRD to improve the nutritional status of patients and prevent diarrhea and dehydration in the long term [13]. In this study, we present for the first time the impact of this protocol of nutritional support in a population of patients with cancer and OTRD.

## 2. Materials and Methods

### 2.1. Study Design

This multicenter, observational, prospective cohort study was conducted at 15 Spanish centers with the involvement of Medical Oncology, Radiation Oncology, and Nutrition departments. The study adhered to the tenets of the Declaration of Helsinki and was approved by the Ethics Committee of Comunidad Autónoma de Aragón (CEICA, Code CP-CI.PI 15/0319). All participants provided written informed consent before inclusion in the study.

The nutritional status of patients, stool characteristics, and other clinical variables were collected before the initiation of nutritional treatment with OEN (baseline) and after 8 weeks. Nutritional support consisted of Survimed OPD Drink^®^ (Fresenius Kabi, Bad Homburg, Germany), a peptide-based sip feed with the following composition: hydrolyzed proteins (18.6%, whey proteins), carbohydrates (56.4%: 30% saccharose and 70% oligosaccharides), and fat content (25%; 51% as medium-chain triglycerides) at a final caloric density of 1 Kcal/mL (Supplementary Information).

The nutritional protocol comprised the following steps: (1) Assessing the nutritional status of patients with the Subjective Global Assessment (SGA), and (2) Evaluating the intestinal function of patients [13] (see Graphical Abstract). Patients received 2 or 3 bottles per day (200 mL/bottle) of OEN according to their nutritional status and intestinal function.

The OEN protocol was started at the onset of OTRD. Furthermore, patients were recommended to follow an astringent diet (low in insoluble fiber and fat, free of lactose, irritating or flatulent foods). Only patients with watery stools received loperamide at baseline until the consistency improved to mushy.

### 2.2. Study Population

Patients were included according to the following criteria: adult patients with confirmed diagnosis of cancer, receiving cancer therapy (antitarget therapy, chemotherapy, radiotherapy, or chemotherapy and radiotherapy), and with malnourishment or at risk of malnourishment associated with OTRD.

Patients were excluded from the study when they did not provide written informed consent, their life expectancy was <3 months, diarrhea was secondary to the treatment with antibiotics, H2-receptor antagonists or prokinetics, laxatives or osmotically active agents, patients with *Clostridium difficile* infection, and with any concomitant gastrointestinal disorder not associated with the tumor or cancer treatment.

### 2.3. Study Outcomes

This study aimed to assess the effectiveness of the proposed OEN protocol in improving the nutritional status and stool frequency and consistency in patients with OTRD.

The nutritional status was assessed at baseline and following 8 weeks of OEN treatment through the SGA [15]. The SGA is a nutritional assessment method that considers body weight, body mass index (BMI), dietary intake, gastrointestinal symptoms, physical examination (loss of subcutaneous fat, muscle wasting, etc.), muscular function, and albumin levels. Patients were classified as: (1) at risk of malnourishment, (2) moderate malnourished or (3) severe malnourished. Based on inclusion criteria, all the patients included had some level of malnourishment, and none was well-nourished, being the category “at risk of malnutrition” the best possible clinical status in these patients. We defined that the protocol was effective when: (1) the nutritional status was maintained at risk of malnourishment, (2) improved from moderate to at risk of malnourishment or from severe to moderate malnourishment (1-level improvement), or (3) improved from severe to at risk of malnourishment (2-level improvement). Worsening of nutritional status was considered when: (1) the nutritional status worsened from at risk to moderate malnourishment or from moderate to severe malnourishment (1-level worsening) or (2) from low risk to severe malnourishment (2-level worsening).

Stool consistency was evaluated with the Bristol Stool form scale [16] at baseline and after 8 weeks of OEN treatment. The Bristol Stool form scale classifies the consistency of stools into 7 types; types 1–4 are considered “normal” and types 5 (soft), 6 (mushy) and 7 (watery) indicate diarrhea. The treatment was considered effective when it (1) allowed the maintenance of normal stools (types 1–4); (2) improved stool consistency by 1 level (from 7 to 6, 6 to 5, or 5 to 1–4 types), 2 levels (from 7 to 5 or 6 to 1–4 types), or 3 levels (from 7 to 1–4 types). Conversely, we considered that stool consistency worsened when stool classification increased by 1 level (from 1–4 to 5, 5 to 6, or 6 to 7 type) or 2 levels (from 1–4 to 6 or 5 to 7).

Stool frequency was evaluated by recording the number of stools per day at baseline and after 8 weeks of OEN treatment. Given the wide inter-patient variability, stool frequency was not categorized and was then expressed as the mean number of stools/day.

Patients reported the average OEN volume consumed throughout the study and compliance was categorized into the following groups: consumption of all (200 mL/bottle) the OEN content prescribed, 2/3 (150 mL/bottle) of the content prescribed, and 1/2 (100 mL/bottle) of the content prescribed.

Tolerance to the OEN treatment was monitored by recording the frequency 2 h after the OEN intake of the following events: nausea, vomiting, reflux, abdominal pain, flatulence, satiety, constipation, or stomach heaviness. Patients rated the frequency of each event as “never”, “rarely”, “sometimes”, “often”, and “always” after 8 weeks of OEN treatment.

### 2.4. Statistical Analyses

The statistical package IBM SPSS Statistics v.24.0 (IBM Corp, Armonk, NY, USA) was used for statistical analysis. Normal distribution of the data was tested with the Kolmogorov–Shapiro–Wilks test and normality plots. Categorical variables were reported as counts and percentages. Continuous variables were reported as mean and standard deviations. Differences between periods in categorical variables were estimated using the McNemar test. Continuous variables were compared across the 2 periods using paired *t*-tests. A *p* < 0.05 was considered statistically significant.

Sample size calculations were performed with the McNemar formula for 2 proportions with paired data. Considering a nutritional improvement in 20% of the population and, assuming a 5% type I (α) error, 90% power, and 20% drop out rate, 145 patients were required for this study.

## 3. Results

### 3.1. Study Population

A total of 162 patients with OTRD who initiated treatment with OEN were included and 149 completed the study. None of the patients received previous enteral nutrition.Mean ± SD age in the overall population was 68.6 ± 12.6 years, and 55% were men. The tumor was localized in 52.9% of cases, and 50.4% of patients received curative treatment, and chemotherapy + radiotherapy was the treatment of choice in 46.3% of cases (Table 1). No patient required a reduction in cancer therapy throughout the study.

### 3.2. Nutritional Status

The nutritional status at baseline assessed with the SGA showed that 34.9% of patients were at risk of malnourishment (lowest level of malnourishment), 54.4% were moderately malnourished, and 10.7% severely malnourished. After 8 of weeks of nutritional support with OEN, the proportion of patients at risk of malnourishment increased (62.2%) while that of moderately and severely malnourished decreased (34.5% and 3.4%, respectively). The changes in the nutritional status from baseline to week 8 were statistically significant (*p* < 0.001). This resulted in an improvement in the nutritional status in 48.3% of patients and a worsening in 14.1% of patients. Taken together, OEN support was effective in 68.5% of cases: 48.3% improved the nutritional status, and 20.1% were maintained at low risk of malnourishment (the best possible status in these patients) (Figure 1).

The nutritional status significantly improved in all patient subgroups regardless of the type of tumor, type of treatment, treatment modality, resectability, and whether they received targeted therapy or cytotoxic treatment (Appendix A).

BMI was significantly higher following 8 weeks of OEN treatment compared to baseline (22.93 vs. 22.64 kg/m^2^; *p* = 0.004) (Figure 2). Serum albumin (g/dL) levels were 2.88 g/dL at baseline and significantly increased with the OEN protocol after 8 weeks (3.45 g/dL; *p* < 0.001) (Figure 2). Differences in the change in BMI did not reach statistical significance for patients receiving radiotherapy, radiotherapy + chemotherapy, targeted therapies, or non-cytotoxic therapy. The change in albumin levels was statistically significant for all the patient subgroups.

### 3.3. Stool Consistency and Frequency

Stool consistency significantly changed with the 8-week OEN protocol (*p* < 0.001). The proportion of patients with normal and soft stools increased (from 14.1% to 33.6% and from 21.5% to 37.6%), whereas mushy and watery stools decreased (from 38.3% to 26.8% and from 26.2 to 2%) (Figure 3). Changes in stool consistency from baseline to week 8 were statistically significant (*p* < 0.001). After 8 weeks of nutritional support with OEN, 57.7% of patients changed the stool consistency, with 24.2% and 24.8% showing 1- and 2-level improvement, respectively. The treatment showed a 71.1% effectiveness in improving stool consistency (13.4% maintained stool types 1–4 and 57.7% improved stool consistency). Stool consistency was significantly better after 8 weeks of OEN treatment in all stratification groups, except for the subgroups of patients receiving targeted therapy or non-cytotoxic treatment.

The mean number of stools per day decreased from baseline (4.17 ± 3.33 stools/day) to week 8 (1.42 ± 1.63 stools/day; *p* = 0.0041) (Figure 4). Stool frequency significantly improved in all stratification groups with the exception of patients that received targeted therapy.

Cross-tabulation of nutritional status levels and stool characteristics showed that the change in nutritional status did not depend on the change in stool consistency (*p* = 0.34; Chi-square).

### 3.4. Patients with Persistent Diarrhea

We next analyzed the subgroup of patients for whom the presence of diarrhea did not improve with the OEN protocol. Despite not improving the consistency or frequency of stools, we observed a significant improvement in the nutritional status (*p* = 0.04) and albumin levels (*p* = 0.009) in those patients with persistent diarrhea. Of note, 46% of patients improved the nutritional status (40% by 1 level and 6% by 2 levels). In those patients with persistent diarrhea, the nutritional status worsened in 16% of patients and was maintained in 38% (with 12% of patients at risk of malnourishment and 26% with moderate malnourishment) (Figure 5).

### 3.5. Compliance

72.3% of patients reported consuming the total content of OEN prescribed, 20.9% 2/3 of the content prescribed and 6.8% ½ of the content prescribed throughout the study. Nutritional support with OEN was associated with good tolerability in 80.3% of patients and with moderate intolerance in 19.7% of patients (Table 2).

When analyzing compliance rates in different patient subgroups, we observed that compliance rates were significantly higher after 8 weeks of OEN treatment relative to baseline in all subgroups. Nevertheless, we found poorer compliance rates in patients under palliative vs. curative care and with unresectable vs. localized tumors. Regarding the modality of cancer treatment, compliance was lower in patients receiving chemotherapy vs. radiotherapy or both, targeted vs. non-targeted therapy, or cytotoxic vs. non-cytotoxic treatment.

The majority of patients rated the frequency of different symptoms as “never” or “rarely” present. Regarding potential symptoms associated with OEN treatment, the reported absence of symptoms was the following: no nausea in 82.3% of patients, no vomiting in 88.4% of patients, no reflux in 78.2% of patients, no flatulence in 69.2% of patients, no abdominal pain in 77.8% of patients, no satiety in 51.7% of patients, no constipation in 91.2% of patients, and no heaviness in 68% of patients (Appendix A).

## 4. Discussion

This study showed that the proposed OEN algorithm seemed effective in improving the nutritional status, frequency and consistency of stools and was associated with high compliance rates in patients with OTRD.

Unlike previous studies, we included 149 oncologic patients with several types of tumors and receiving different cancer treatment modalities. The overall analysis of patients allowed us to assess the general response to OEN in OTRD patients, whereas patient subgroups were useful for assessing specific responses. In the overall population, mean age was 68.6 years, most of the patients included were at risk of malnourishment or moderately malnourished (88.4%), and the most common tumor type was colon.

An important finding of our study is the improvement in the nutritional status observed in 48.3% of patients and the effectiveness of the treatment (68.5%). We observed an increase in the proportion of patients at risk of malnourishment and a decrease in those with moderate/severe malnourishment in the overall population and in different patient subgroups in our study. Unlike other studies using the turnover rate of plasma proteins (prealbumin, transferrin, and retinol-binding protein) and changes in body weight as indicators of nutritional status, this study comprehensively assessed the nutritional status of patients through the SGA, BMI change, and albumin levels. Indeed, to our knowledge, none of the studies using elemental diet in cancer patients used the SGA to assess the nutritional status, despite the fact that it is considered the recommended nutritional assessment test for cancer patients. The randomized controlled trial comparing the efficacy of oligomeric vs. polymeric formulas used the Prognostic Nutritional Index (PNI) and the Controlling Nutritional Status (CONUT) and showed significantly better scores for the CONUT in patients receiving the oligomeric formula [17]. Most of the studies using elemental diet formulations reported no significant differences in the nutritional status between intervention and control groups in cancer patients [18,19,20,21]. In contrast, in one study the elemental diet improved the intestinal mucosal inflammation and nutritional status of patients with Crohn’s Disease [14]. However, it is important to note that previous results were based on well-nourished patients at baseline, and in our study, all the patients included had certain level of malnourishment.

In line with the improvement in nutritional status revealed by the SGA, serum albumin levels significantly increased from 2.88 to 3.45 g/dL upon treatment with OEN. A previous study reported a decrease in albumin after 4 weeks of Elental^®^ nutritional support (80 g/300 kcal amino-acid-rich, fat free, elemental diet) [22], while in other study the changes were minimal and non-significant after 6 months of either oligomeric or polymeric formula [17] or 14 days of microbial immune nutrition [23].

OEN was also associated in our study with a significant increase in BMI after 8 weeks. Previous results reporting the effect of nutritional support on BMI have been heterogeneous. The effect of Elental^®^ on body weight was inconsistent, with one randomized study reporting a significant difference in body weight change when compared with the control group in patients who received adjuvant chemotherapy for gastric cancer [24], other study showing a preservation of lean body mass after 4 weeks of nutritional support during chemo- or chemoradiotherapy in patients with esophageal cancer [22], and two studies showing no differences in patients with oral squamous cell carcinoma [18,19]. Furthermore, Ohkura et al. found no significant differences in the change in body weight between oligomeric and polymeric formulas after esophagectomy [17]. The importance of body weight improvement is derived from the association between this variable and negative quality of life and worse long-term prognosis [25] and also considering that severe weight loss (>10%) has been acknowledged as a risk factor for non-relapse mortality [26].

The OEN protocol was effective in improving stool consistency, with 57.7% of patients showing an improvement, and a remarkable reduction in the proportion of watery stools (from 25.5% to 2%). These improvements resulted in the OEN treatment being effective for stool consistency in 71.1% of cases. In analyzing these results, it is important to consider that patients with watery stools at baseline received loperamide until changing the consistency to mushy. Although the use of this antidiarrheal drug contributed to improving the consistency in those patients, its use in only 26.2% of patients does not fully explain the improvement in stool consistency observed in the overall population (57.7%). Likewise, the reduction in stool frequency from 4.17 stools/day at baseline to 1.42 stools/day after 8 weeks was also striking. Regarding cancer therapy modality, the higher stool frequency per day at baseline was observed in patients receiving chemotherapy and radiotherapy (5.46 stools/day) as previously reported [8], although this subgroup also experienced the highest decrease in stool frequency after OEN (reduction of −4.13 stools/day). In this context, it is interesting to note that the subgroup of patients receiving both chemotherapy and radiotherapy also showed the highest compliance. Regarding previously reported effects of enteral nutrition on diarrhea, a study found that the frequency of diarrhea was significantly lower with an oligomeric vs. a polymeric formula [17], and two showed a higher incidence of diarrhea in the elemental diet group as compared with the control group [27] or azulene oral rinse [22].

A relevant finding of this study is that the OEN support seemed effective in improving the nutritional status even in those patients with persistent diarrhea despite the OEN protocol. This result is of remarkable interest since it implies that patients could benefit from this OEN protocol, despite the fact that they do not notice substantial changes in the frequency and consistency of stools.

Patients reported high compliance rates with the OEN support, with 72.3% reporting ingesting the total content prescribed. The high compliance rates observed are in line with the good tolerability (80.3%) associated with this nutritional support. Importantly, common symptoms responsible for enteral nutrition interruptions [12], such as nausea or vomiting, were seldom present in our study. Satiety, heaviness and flatulence were the symptoms with a higher incidence, although still in >80% of cases were “never” or “rarely” reported. These results are of particular importance, as poor tolerability and palatability of oral nutritional formulas, together with tolerability symptoms associated with cancer treatment, are the main barriers to treatment compliance that lead to suboptimal effectiveness of nutritional supports [28]. Previous studies provided heterogeneous results regarding treatment compliance. In patients receiving chemotherapy and Elental^®^ formula as elemental diet, a compliance rate of 71.4% was reported for patients with gastric cancer [24] and of 70% in patients with esophageal cancer [29]. Poorer compliance rates were observed in patients undergoing pelvic radiotherapy and partial replacement with oral E028, showing a median overall daily intake of 21% of caloric requirements [21].

Taken together, we observed that OEN seemed to serve its first purpose of improving the nutritional status of patients. One of the main contributors to this effectiveness could be the compliance with the treatment that allows the optimal effect of this nutritional support. Treatment compliance, in turn, is likely influenced by the improvement in health status (nutritional, body weight, frequency of stools), which further encourages patients to remain compliant. The improved nutritional status is also probably reflecting the higher digestibility of a peptide-based formulation, which has the potential to maintain the gastrointestinal mucosa integrity and facilitate nutrient absorption [29]. Furthermore, oligomeric formulas have been associated with reduced mucosal production of pro-inflammatory cytokines [14] and pancreatic or biliary stimulation [30]. It is important to highlight that we cannot directly compare our results with those using elemental supplements previously published given their different methodological design [13]. First, most of the studies included a limited number of patients [22,29,31]. Second, the nutritional doses provided were different (in most of the studies patients received 300 Kcal and 14.1 g/protein/day and, in our study, patients received between 400 Kcal and 18.6 g/protein/day and 600 Kcal and 27.9 g/protein/day) [18,29,32]. Third, the follow-up period of these studies was relatively short for achieving nutritional changes [21,22]. Fourth, none of the studies primarily aimed to assess the impact of nutritional supports on the nutritional status and, in several of them, this was neither a secondary endpoint [31]. Fifth, when reported, patients were well-nourished at baseline, so better results are expected in patients with certain level of malnourishment as in our study [18,19,22,24].

There are different limitations associated with this study that are worth mentioning. First, the lack of control group does not allow to rule out that factors, other than the OEN protocol, could have conditioned the results obtained. Potential factors could be following an astringent diet in all patients, the use of transient antidiarrheal drugs in patients with watery stools until changing the consistency to mushy, or the evolution of diarrhea itself. Further investigations with a controlled design are needed to confirm our results. However, as this is the first study following the proposed OEN protocol in clinical practice, our study was not intended to compare the effectiveness of this protocol with other nutritional support but to test whether the protocol help improve the nutritional status. In this regard, since this observational study included patients with nutritional deficit for whom the absence of nutritional support could have compromised their status, the use of placebo was not considered. Moreover, as previous animal studies reported reduced pancreatic or biliary stimulation with oligomeric vs. polymeric diet [30] and a lower frequency of diarrhea and improved nutritional status was observed with a oligomeric vs. a polymeric formula [17], we did not consider comparing the effectiveness of our OEN protocol with a polymeric formula. Second, the inclusion of patients did not follow selective inclusion criteria. Although this allowed us to obtain comprehensive information among a wide range of clinical groups, the sample size in some subgroups was not powered to establish reliable conclusions. Third, the duration of follow up was relatively short to capture long term differences. Fourth, we cannot directly compare the compliance rates obtained in our study with those previously observed, as patients reported the total volume ingested throughout the study and other studies were based on the doses recorded in the patient diary. Fifth, antidiarrheal drugs were used only in patients with watery stools until the consistency improved to mushy. As the proportion of patients who reported watery stools at baseline was 26.2%, the use of loperamide does not completely explain the improvement in the frequency and consistency of stools observed in the overall population. Although loperamide undoubtedly contributed to improving the consistency of stools, we observe this effect also in patients not treated with this antidiarrheal drug (soft or mushy stools consistency, 59.8% of patients). Finally, due to the observational design of our study, we cannot ensure that the clinical benefits observed are fully due to the OEN protocol.

In contrast, this study is one of the first comprehensively describing the effectiveness of an OEN protocol targeting both the nutritional status of patients and the evolution of OTRD. Our study included a considerable sample size, which is larger than that in most of the studies published to date and assessed the nutritional status with a valid and reliable tool [33,34] and with other nutritional indicators (albumin, BMI). To our knowledge, this study is the first in assessing the change in nutritional status with an OEN protocol in patients with different types of tumors. Moreover, most of the previous studies were conducted in Japan [17,18,19,20,22,24,27,29,31], and the present study provides valuable information for patients in Europe.

## 5. Conclusions

The OEN protocol proposed in this study seemed effective in improving patient’s nutritional status, stool frequency, and consistency with high rates of compliance in patients with OTRD.

## Figures and Tables

**Figure 1 nutrients-12-01534-f001:**
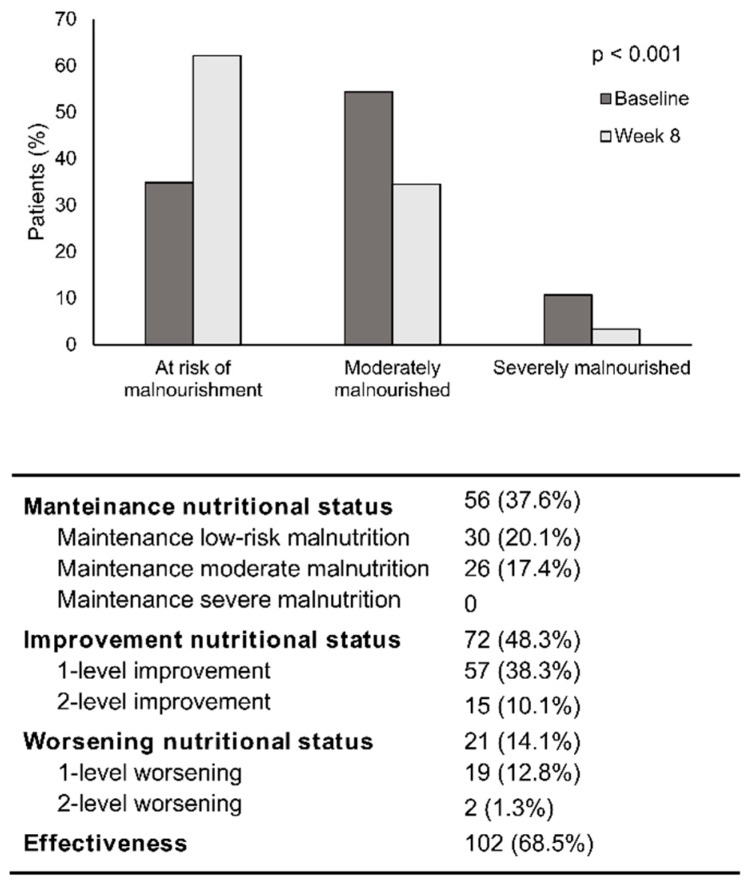
Change in nutritional status after 8 weeks of oligomeric enteral nutrition (OEN). The bars show the proportion of patients at risk of malnourishment and with moderate or severe malnourishment at baseline and after 8 weeks of OEN treatment. The change in the nutritional status from baseline to week 8 was statistically significant (*p* < 0.001). The table shows the changes in the nutritional status (improvement and worsening) and the effectiveness of the OEN algorithm in improving the nutritional status. Patient’s nutritional status was classified as: (1) at risk of malnourishment, (2) moderate malnourishment, or (3) severe malnourishment.

**Figure 2 nutrients-12-01534-f002:**
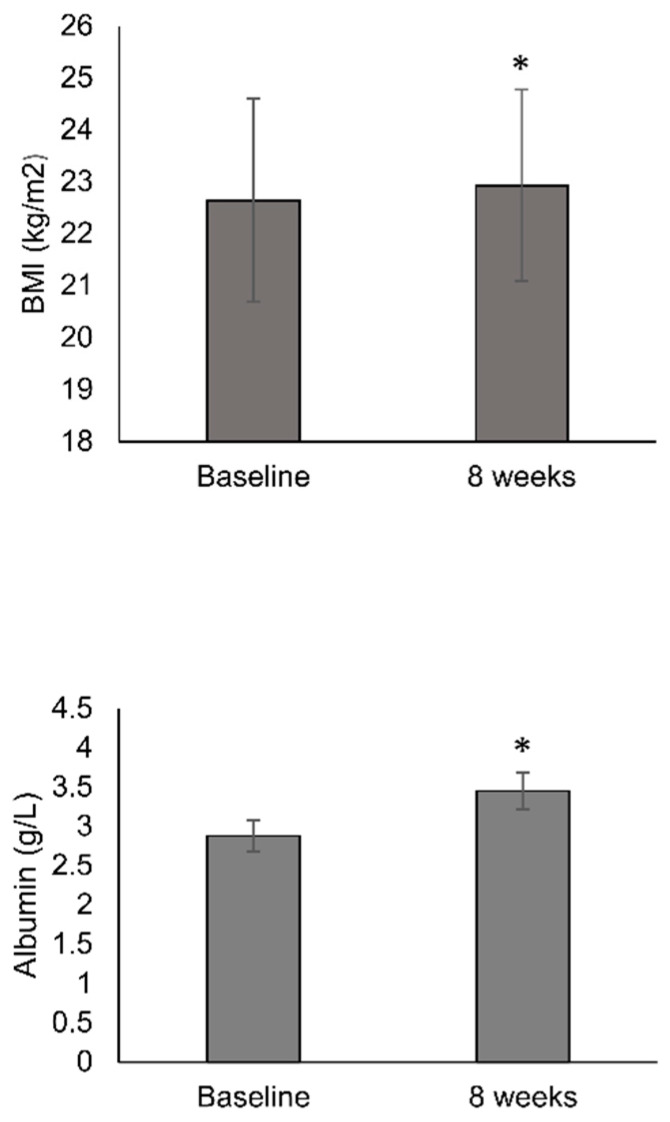
Change in body mass index and albumin levels after 8 weeks of oligomeric enteral nutrition (OEN). The bars show mean ±SD levels for body mass index (BMI) and albumin at baseline and after 8 weeks of OEN treatment. * indicates *p* < 0.05 (from baseline to week 8).

**Figure 3 nutrients-12-01534-f003:**
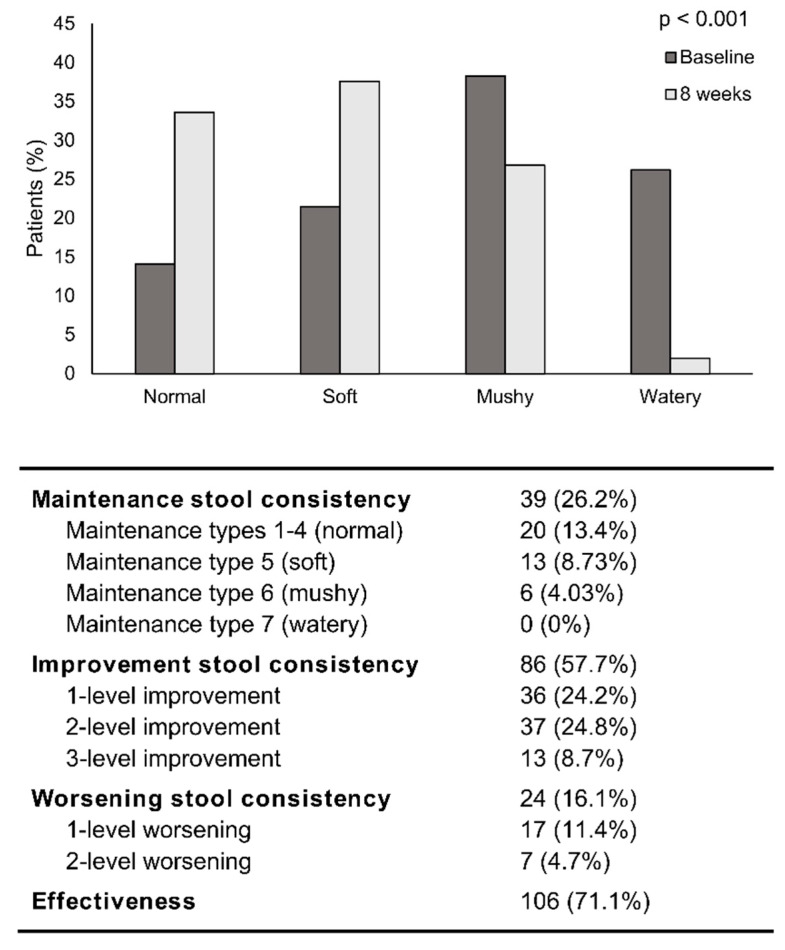
Change in stool consistency after 8 weeks of oligomeric enteral nutrition (OEN). The bars show the proportion of normal, soft, mushy, and watery stools at baseline and after 8 weeks of OEN treatment. Changes in stool consistency from baseline to week 8 were statistically significant (*p* < 0.001). The table shows the proportion of patients with changes in stool consistency and the effectiveness of the OEN protocol in improving stool consistency.

**Figure 4 nutrients-12-01534-f004:**
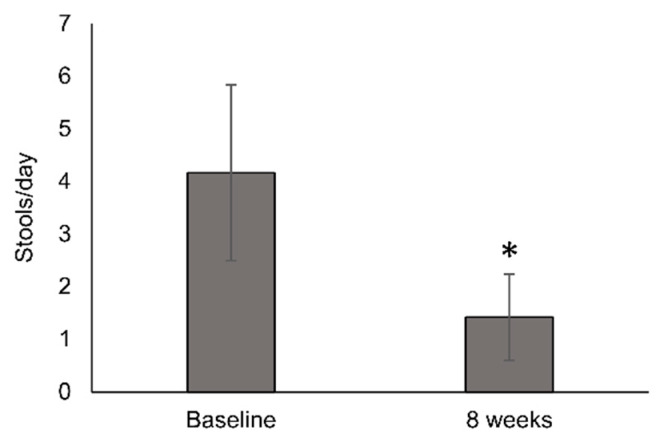
Change in stool frequency after 8 weeks of oligomeric enteral nutrition (OEN) treatment. The bars show mean ± SD stools/day at baseline and after 8 weeks of OEN treatment. The change in stool frequency from baseline to week 8 was statistically significant (*p* = 0.0041). * mean the statistical significance (*p* = 0.0041).

**Figure 5 nutrients-12-01534-f005:**
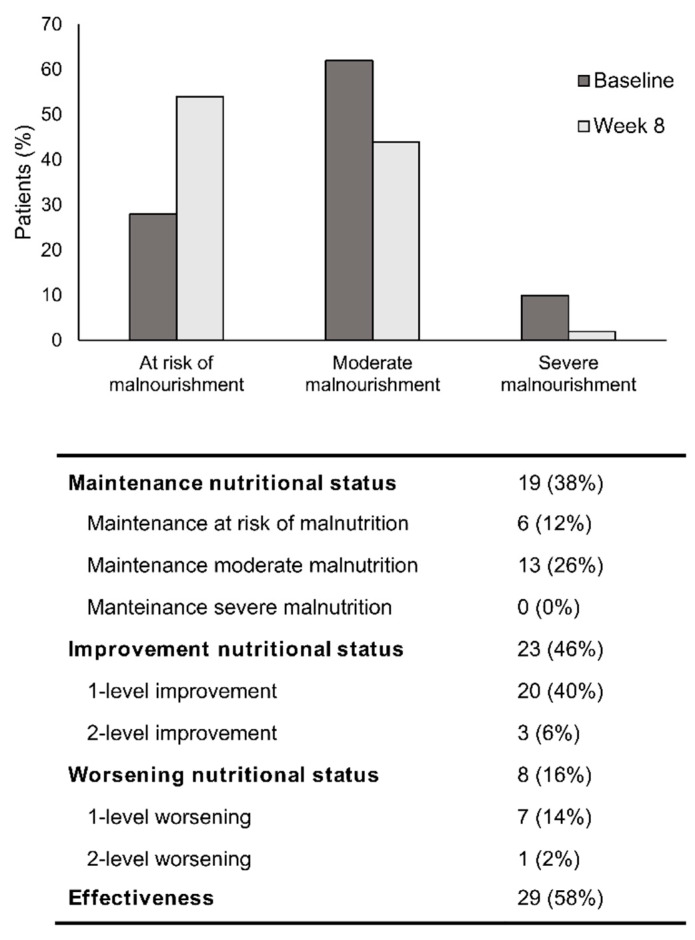
Nutritional status evolution in those 50 patients who did not improve diarrhea after 8 weeks of oligomeric enteral nutrition (OEN). The graph shows the proportion of patients at risk of malnourishment, with moderate or severe malnourishment at baseline and after 8 weeks of nutritional support with OEN. The table shows the proportion of patients with changes in the nutritional status and the effectiveness of the OEN protocol in improving the nutritional status in patients with persistent diarrhea.

**Table 1 nutrients-12-01534-t001:** Demographic and clinical characteristics of study participants.

	*n* = 149
Age (years), Mean (SD)	68.6 (12.6)
Median (min; max)	69 (30; 92)
Gender, (%)	
Men	82 (55.0%)
Woman	67 (45.0%)
Weight (kg), Mean (SD)	62.9 (12.1)
BMI (kg/m^2^), Mean (SD)	22.6 (3.9)
Resectability (* *n* = 136)	
Unresectable	64 (47.1%)
Localized	72 (52.9%)
Type of treatment (* *n* = 131)	
Palliative	65 (49.6%)
Curative	66 (50.4%)
Treatment modality	
Chemotherapy	56 (37.6%)
Radiotherapy	24 (16.1%)
Chemotherapy + Radiotherapy	69 (46.3%)
Targeted therapy	
Yes	28 (18.8%)
No	121 (81.2%)
Cytotoxic treatment	
Yes	99 (66.4%)
No	50 (33.6%)
Type of tumor	
Gynecologic/urologic	29 (19.5%)
Colon	78 (52.3%)
Esophagogastric	22 (14.8%)
Other	20 (13.4%)

* number of patients with available data. % are calculated over the patients with available data. BMI, Body Mass Index. Demographic and clinical characteristics were not significantly different by sex, except for the proportion of specific types of tumors (such as gynecologic).

**Table 2 nutrients-12-01534-t002:** Compliance and tolerance to oligomeric enteral nutrition (OEN) after 8 weeks of treatment.

Treatment Compliance (* *n* = 148)	
Total content prescribed	107 (72.3%)
2/3 bottle content prescribed	31 (20.9%)
1/2 bottle content prescribed	10 (6.8%)
Treatment tolerance (**n* = 147)	
Good tolerance	118 (80.3%)
Moderate intolerance	29 (19.7%)
Severe intolerance	0 (0.0%)

* Number of patients with available data. % are calculated over the patients with available data.

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
