# Peer review of "Clinical and Nutritional Effectiveness of a Nutritional Protocol with Oligomeric Enteral Nutrition in Patients with Oncology Treatment-Related Diarrhea"

_nutrients, 2020, doi:10.3390/nu12051534_

Round 1

Reviewer 1 Report

This investigation examined the use of an oligomeric enteral nutrition supplement for patients with oncology treatment related diarrhea over an 8 week period. They showed an improvement in nutrition status for those with moderate and severe malnourishment and an improvement in diarrhea. However, I have some concerns and suggestions for improvement of the manuscript that the authors need to address.

A major problem with this study is the lack of a control group which the authors discussed. How do we know that these improvements occurred because of the diet versus the natural progression of the disease process?

How is this supplement different from other elemental supplements? What is unique about it?

Did the patients fail tolerance of a polymeric supplement? The authors suggest the polymeric formulas are discouraged. Is there a sufficient amount evidence for their failure? How was failure defined or were patients empirically given the OEN without consideration of a polymeric diet?

In addition to the diet, what other dietary intakes were allowed/encouraged?

What anti-diarrheal medications were given during the 8 week period? Could the improvement in diarrhea be related to medications, disease process, or diet? Although lack of consideration of anti-diarrheal medication was brought up in the limitations, we really need to have that information to assess the efficacy of this supplementation. We will never know if the improvement in diarrhea was attributed to the diet versus the medication without it.

After 8 weeks of nutrition support with the OEN, the proportion of patients at risk for malnourishment increased. Why? Uncontrolled diarrhea?

When was this diet implemented relative to their cancer therapy? At the point of developing diarrhea? At the outset of therapy? Please explain in the methods.

Line 277. The authors state most studies with elemental formulas did not improve nutrition status. So why did this one work? Was it the formula or other procedures complementing the nutrition therapy. The type of cancers (where it failed) versus this population? The type of oncology therapy given? Please explain.

Author Response

REVIEWER 1

This investigation examined the use of an oligomeric enteral nutrition supplement for patients with oncology treatment related diarrhea over an 8 week period. They showed an improvement in nutrition status for those with moderate and severe malnourishment and an improvement in diarrhea. However, I have some concerns and suggestions for improvement of the manuscript that the authors need to address.

We thank the reviewer for his/her work and comments that significantly improved the quality of the manuscript. Sometimes, the authors do not notice that some parts of the work are not clear, being of great value the suggestions of the reviewers that make the work more understandable for future readers and clear up questions that would otherwise remain unresolved.

A major problem with this study is the lack of a control group which the authors discussed. How do we know that these improvements occurred because of the diet versus the natural progression of the disease process?

As rightly pointed by the reviewer, and already mentioned in the limitations of the Discussion, the lack of control group is an important limitation of the study. To further reinforce the importance of this limitation, we have added the following information to the new version of the manuscript (pages 13-14, line 366-376):

“There are different limitations associated with this study that are worth mentioning. First, the lack of control group does not allow to rule out that factors, other than the OEN protocol, could have conditioned the results obtained. Potential factors could be following an astringent diet in all patients, the use of transient antidiarrheal drugs in patients with watery stools until changing the consistency to mushy, or the evolution of the diarrhea itself. Further investigations with a controlled design are needed to confirm our results. However, as this is the first study following the proposed OEN protocol in clinical practice, our study was not intended to compare the effectiveness of this protocol with other nutritional support but to test whether the protocol help improve the nutritional status. In this regard, since this observational study included patients with nutritional deficit for whom the absence of nutritional support could have compromised their status, the use of placebo was not considered.”

We have also included limitations regarding the use of antidiarrheal drugs or that the observational design of the study does not allow to ensure that clinical benefits are fully due to the OEN protocol (page 14, lines 389-396).

How is this supplement different from other elemental supplements? What is unique about it?

We agree with the reviewer that this issue needed clarification and thank the suggestion. Accordingly, we have included the following information (page 13, lines 355-365):

It is important to highlight that we cannot directly compare our results with those using elemental supplements previously published given their different methodological design (Sanz et al. 2019). First, most of the studies included a limited number of patients (Ogata et al. 2016, Tanaka et al. 2018, Ishikawa et al. 2016). Second, the nutritional doses provided were different (in most of the studies patients received 300 Kcal and 14.1 g/protein/day and, in our study, patients received between 400 Kcal and 18.6 g/protein/day and 600 Kcal and 27.9 g/protein/day) (Tanaka et al. 2016, Ogata et al. 2016, Harada et al. 2018). Third, the follow-up period of these studies is relatively short for achieving nutritional changes (McGough et al. 2004, Ishikawa et al. 2016). Fourth, none of the studies primarily aimed to assess the impact of nutritional supports on the nutritional status and, in several of them, this was neither a secondary endpoint (Ogata et al. 2016). Fifth, when reported, patients were well-nourished at baseline, so better results are expected in patients with certain level of malnourishment as in our study (Ishikara et al. 2016, Harada et al. 2016, Harada et al 2018, Toyomasu et al. 2019).

Did the patients fail tolerance of a polymeric supplement? The authors suggest the polymeric formulas are discouraged. Is there a sufficient amount evidence for their failure? How was failure defined or were patients empirically given the OEN without consideration of a polymeric diet?

We thank the reviewer for this interesting clarification.

None of the patients previously received enteral nutrition. We have included this information in the Results (page 4, line 154).

In the review published by our group in Nutrients (Sanz et al. 2019) we already pointed out to the more favorable effects of oligomeric vs polymeric enteral nutrition in cancer patients with diarrhea. This was based on animal studies reporting reduced pancreatic or biliary stimulation with oligomeric vs regular or polymeric diet (O'Keefe et al. 2003) and the reduced frequency of diarrhea and improved nutritional status with an oligomeric formula vs a polymeric one (Ohkura et al. 2019). Accordingly, the present protocol is based on the use of OEN. We have included this evidence in the new version of the manuscript (page 14, lines 376-379).

In addition to the diet, what other dietary intakes were allowed/encouraged?

We thank the reviewer for this relevant question that is necessary to understand the results. The OEN treatment was initiated at the onset of OTRD and for all the patients astringent diet was recommended.

We have included this information in the Study design section (page 2, lines 89-92): “The OEN protocol was started at the onset of OTRD. Furthermore, patients were recommended to follow an astringent diet (low in insoluble fiber and fat, free of lactose, irritating or flatulent foods).”

In addition, in the limitations of the Discussion section (page 14, lines 368-370) we mentioned that following an astringent diet could have conditioned the results obtained.

What anti-diarrheal medications were given during the 8 week period? Could the improvement in diarrhea be related to medications, disease process, or diet? Although lack of consideration of anti-diarrheal medication was brought up in the limitations, we really need to have that information to assess the efficacy of this supplementation. We will never know if the improvement in diarrhea was attributed to the diet versus the medication without it.

We agree with the reviewer that the use of antidiarrheal medication is an important consideration to analyze the results.

Throughout the study, loperamide was prescribed in patients with watery stools until the consistency improved to mushy. The proportion of patients who reported watery stools at baseline was 26.2%, so the use of loperamide does not completely explain the improvement in the frequency and consistency of stools observed in the overall population. Although the use of loperamide likely improved stool consistency and frequency, we observed an improvement in the consistency of stools in patients not treated with this antidiarrheal drug (soft or mushy stools consistency, 59.8% of patients). We have included this relevant information in the manuscript thanks to the reviewer suggestion (page 14, lines 389-395). Moreover, is it important to note that none of the patients reduced the oncological treatment, so no bias is expected from this factor.

After 8 weeks of nutrition support with the OEN, the proportion of patients at risk for malnourishment increased. Why? Uncontrolled diarrhea?

We agree with the reviewer that this concept was not clear in the previous version of the manuscript. Patients were classified based on their nutritional status in: 1) at risk of malnourishment, 2) moderate malnourished or 3) severe malnourished. Based on inclusion criteria, all the patients included had some level of malnourishment and was well-nourished, being the category “at risk of malnutrition” the best clinical status in these patients. To clarify this point and avoid giving misleading information we have included this information in  Study outcomes section (page 3, lines 111-115): 

In addition, we have added in the Results (Nutritional status section, page 5, line 173) “Taken together, OEN support was effective in 68.5% of cases: 48.3% improved the nutritional status and 20.1% were maintained at low risk of malnourishment (the best possible status in these patients)”.

When was this diet implemented relative to their cancer therapy? At the point of developing diarrhea? At the outset of therapy? Please explain in the methods.

This is an important point to review in the Material and Methods section. This OEN protocol was implemented at the onset of OTRD, together with the recommendation of following an astringent diet. As reported in the Inclusion criteria, patients included were already on oncological treatment at study initiation and the start of the OEN protocol depended on the onset of OTRD.

Accordingly, we have added in Study design section (page 2, lines 89-91) the following statement: “The OEN protocol was started at the onset of OTRD. Furthermore, patients were recommended to follow an astringent diet (low in insoluble fiber and fat, free of lactose, irritating or flatulent foods).”

Line 277. The authors state most studies with elemental formulas did not improve nutrition status. So why did this one work? Was it the formula or other procedures complementing the nutrition therapy. The type of cancers (where it failed) versus this population? The type of oncology therapy given? Please explain.

As already mentioned in commentary 3 and, included in the new version of the manuscript (page 13, line 355-365) several methodological issues impede the direct comparison between our study and other previously published, such as the sample size, primary endpoints, the nutritional content provided or the nutritional status of patients at baseline. One of the factors that likely contributed to the good clinical results observed is that patients at baseline had various levels of malnourishment, contrasting with previously published studies where patients were well-nourished at baseline.

Furthermore, at the end of the section on study limitations (page 14, lines 395-396), we have included: “Finally, due to the observational design of our study we cannot ensure that the clinical benefits observed are fully due to the OEN protocol”.

Reviewer 2 Report

I evaluated the manuscript “Clinical and Nutritional Effectiveness of a Nutritional Protocol with Oligomeric Enteral Nutrition in Patients with Oncology Treatment-Related Diarrhea” by Sanz-Paris et al. This is a multicentre, observational, prospective study that evaluated the effectiveness of an oligomeric enteral nutrition (OEN) protocol in the improvement of nutritional status and reduction of diarrhea symptoms.The proposed OEN protocol seemed to be effective in improving the nutritional status, frequency and consistency of stools in patients with oncology treatment-related diarrhea even in persistent cases.

The topic is very interesting and original, and the paper is very well written. I do not have any concerns or suggestion.

Author Response

REVIEWER 2

I evaluated the manuscript “Clinical and Nutritional Effectiveness of a Nutritional Protocol with Oligomeric Enteral Nutrition in Patients with Oncology Treatment-Related Diarrhea” by Sanz-Paris et al. This is a multicentre, observational, prospective study that evaluated the effectiveness of an oligomeric enteral nutrition (OEN) protocol in the improvement of nutritional status and reduction of diarrhea symptoms.The proposed OEN protocol seemed to be effective in improving the nutritional status, frequency and consistency of stools in patients with oncology treatment-related diarrhea even in persistent cases.

The topic is very interesting and original, and the paper is very well written. I do not have any concerns or suggestion.

We thank the reviewer for reviewing our manuscript and for the positive comments on our work. As the reviewer may know, sometimes studies do not offer us the expected results, but we feel doubly satisfied when the results are exciting and helpful in daily clinical practice.

Reviewer 3 Report

I have some doubts:

1)   Results_3.1. Study population: the authors included 149 patients that completed the study but they declared that complete clinical data were available for 138 patients. In Table 1“ Demographic and clinical characteristics of study participants” are reported 149 patients and not 138 patients. Why?

2)   In my opinion authors should specify if patients have followed specific dietary advices (low fiber and lactose free diet)

3)   It could be useful if the authors could specify how many patients, who had an improvement of stool frequency,  assumed antidiarrheal drugs and/or faced a reduction of dose delivery in the oncologic treatment.

Author Response

REVIEWER 3

We thank the reviewer for his/her work and comments that significantly improved the quality of the manuscript. Sometimes, the authors do not notice that some parts of the work are not clear, being the expertise of the reviewers of great value. These types of suggestions make the work more understandable for future readers and clear up questions that would otherwise remain unresolved.

I have some doubts:

  • 1. Study population: the authors included 149 patients that completed the study but they declared that complete clinical data were available for 138 patients. In Table 1“ Demographic and clinical characteristics of study participants” are reported 149 patients and not 138 patients. Why?

We thank the reviewer for noticing this issue and regret the previous misleading information. The study included 149 patients, but only 138 had data for all the variables included in the protocol at the end of follow-up (8 weeks). Concretely, data on 138 patients were available for the secondary variable “quality of life” that we decided not to include in this manuscript as we aim to publish these data in a 2nd manuscript. As data for the 149 patients were available for all the variables included in the manuscript, we have removed the sentence “complete clinical data were available for 138 patients (page 4, lines 154-155)” since it could be misleading for readers.

2)   In my opinion authors should specify if patients have followed specific dietary advices (low fiber and lactose free diet)

We thank the reviewer for pointing this out, as this information is relevant to understand the results. As indicated in the new version of the manuscript (Study design section, page 2, lines 89-92), “The OEN protocol was started at the onset of OTRD. Furthermore, patients were recommended to follow an astringent diet (low in insoluble fiber and fat, free of lactose, irritating or flatulent foods).”

In addition, in the limitations of the Discussion section (page 14, lines 368-370) we mention that following an astringent diet could have conditioned the results obtained.

3)   It could be useful if the authors could specify how many patients, who had an improvement of stool frequency,  assumed antidiarrheal drugs and/or faced a reduction of dose delivery in the oncologic treatment.

We agree with the reviewer that the use of antidiarrheal medication and the possible reduction of oncologic treatment could have been determinants in the improvement of stool frequency.  

Of note, stool frequency was not categorized as other variables since it presented a high inter-patient variability and there is no standarized categorization for stool frequency in the bibliography. In the previous version of the manuscript it was indicated (page 3, lines 130-132): “Stool frequency was evaluated by recording the number of stools per day at baseline and after 8 weeks of OEN treatment. Given the wide inter-patient variability, stool frequency was not categorized and was then expressed as the mean number of stools/day”.

In addition, loperamide prescription was not based on stool frequency but on stool consistency.

We explain andiarrheal medication use in more detail in the limitations of the Discussion section (page 14, lines 389-395): “Fifth, antidiarrheal drugs were used only in patients with watery stools until the consistency improved to mushy. As the proportion of patients who reported watery stools at baseline was 26.2%, the use of loperamide does not completely explain the improvement in the frequency and consistency of stools observed in the overall population. Although loperamide undoubtedly contributed to improving the consistency of stools, we observe this effect also in patients not treated with this antidiarrheal drug (soft or mushy stools consistency, 59.8% of patients).” We thank the reviewer for this suggestion, as the manuscript is now easier to interpret.

Regarding cancer treatment, no patient required a reduction in cancer therapy throughout the study. We have included this information in the study population section of the Results (page 4, line 158).

Round 2

Reviewer 1 Report

The authors have adequate addressed my concerns.